# Socio-Cultural Adaptation and Its Related Factors for Chinese Medical Aid Team Members (CMATMs) in Africa

**DOI:** 10.3390/ijerph16173132

**Published:** 2019-08-28

**Authors:** Xiaochang Chen, Xiaojun Liu, Zongfu Mao

**Affiliations:** 1School of Health Sciences, Wuhan University, 115# Donghu Road, Wuhan 430071, China; 2School of Humanities and Management, Guangdong Medical University, 1# Xincheng Road, Dongguan 523808, China; 3Institute of Health Law and Policy, Guangdong Medical University, 1# Xincheng Road, Dongguan 523808, China; 4Global Health Institute, Wuhan University, 115# Donghu Road, Wuhan 430071, China

**Keywords:** expatriate, socio-cultural adaptation, Chinese medical aid team, China, Africa

## Abstract

Chinese Medical Aid Team Members (CMATMs) have a significant impact on the implementation of China’s health assistance strategies in Africa. The socio-cultural adaptation of CMATMs not only relates to the psychological situation and life quality of each member of the medical aid team, but also to the health aid performance of every single member and the medical aid team as a whole. This study evaluated CMATMs’ socio-cultural adaptation and its related factors. This was a cross-sectional survey study. The participants’ socio-cultural adaptation was measured by the Chinese version of the Socio-cultural Adaptation Scale (CSCAS). Stepwise multiple linear regression models were used to identify the main factors that are associated with CMATMs’ socio-cultural adaptation in general and in each dimension. The present study demonstrated that, to some extent, CMATMs are facing problems in socio-cultural adaptation, specifically in interaction. The type of service region, annual aid income, and length of service were identified as the main factors that were the most predictive of the CMATMs’ socio-cultural adaptation in Africa. This study obtained baseline information that is useful to relevant stakeholders in their attempts to improve CMATMs’ socio-cultural adaptation.

## 1. Introduction

Over the past few decades, China has been making great effort to participate in global governance, particularly in the field of global health. It is true that providing assistance in improving health services in African countries is one of China’s priorities in this field. In addition, dispatching medical aid teams to Africa is one of the main channels for offering health assistance to African countries, as well as one of the concrete manifestations of China’s active participation in global governance [1,2,3]. From 2000 to 2013, China’s medical aid team programmes in Africa totalled 65.20 million USD, with 189 projects being undertaken [4]. By 2017, China had sent 25,000 Chinese Medical Aid Team Members (CMATMs) to African countries [5], and there are currently still thousands of CMATMs providing services in recipient countries [3,5].

Nevertheless, China has been showing its strong willingness to take on more responsibilities to help African countries in the field of health. At the Forum on China-Africa Cooperation, China pledged that it would continue to send medical aid teams to Africa and expand the scope and scale of the health care cooperation between China and Africa [6,7]. Obviously, CMATMs have a significant impact on the implementation of China’s health assistance strategies in Africa. However, few studies have been found thus far on this particular population.

As expatriates, CMATMs have to face socio-cultural adaptation, which is one dimension of cross-cultural adaptation. Cross-cultural adaption is one of the important factors that affect the success of international assignments, and it has been attracting scholars’ attention, because, while the number and costs of expatriates have been rocketing, many international assignments have failed [8]. Ward and colleagues [9] classified cross-cultural adaption or adjustment into two dimensions: socio-cultural adaptation (the ability to manage day-to-day life in the host culture context) and psychological adaptation (subjective feelings of well-being and satisfaction). These two dimensions are discrete as well as interrelated. Socio-cultural adaptation is assessed based on the amount of social difficulty experienced in the new environment, and is affected by general cultural knowledge, length of residence, and the amount of social contact with host culture; whereas, psychological adaptation is measured based on mood states (e.g., anxiety, depression, and exhaustion), and is affected by personality, life changes, and social support [10]. On the other hand, these two dimensions are closely related [10,11]. Positive sociocultural adaptation enhances psychological health status and psychological well-being promotes an individual’s learning development of intercultural interactions and communication skills [11]. Some studies have categorised the influencing factors for cross-cultural adaption into internal and external factors [12,13,14,15]. The internal factors include personality, cognitive assessment, coping style, culture-related knowledge and skills, and demographic factors; the external factors include life changes, cultural distance, social supports, contact degree, and discrimination and prejudice [16,17].

Socio-cultural adaptation refers to the ability to manage day-to-day life in the host culture context, reflecting the degree of ease at navigating one’s daily life. In other words, socio-cultural adaptation is about daily life, which constitutes the largest quantity of expatriate’s overseas life and time, hence socio-cultural adaptation plays the primary role in cross-cultural adaptation. Being about daily life, socio-cultural adaptation includes the dimensions of living environment, social morality, social service, social support, and interaction [18]. Although these dimensions have different emphasis, they all focus on an individual’s daily life. For example, the dimension of living environment mainly concerns physical surroundings, such as adaptation to residential environment, weather, diet, and traffic, while the dimension of social service adaptation focuses on restaurant and shop services, foreign banking services, and local hospital services, which are frequently used daily service.

The proven importance of socio-cultural adaptation provides proof for the judgement that socio-cultural adaptation plays the primary role in cross-cultural adaptation, as the maladjustment of this dimension leads to psychological withdrawal, poor performance, and early return [19,20]. Those with severe problems adjusting to their new environment can become depressed or even attempt suicide [21], which also proves that socio-cultural adaptation leads to the failure of psychological adaptation. For example, a study found that 80% of the volunteer Chinese language teachers sent to South Africa have experienced psychological problems, including loneliness (36%), helplessness (18%), irritability (12%), and depression (14%) [22]. Moreover, stressors in one domain can influence stresses in another [20,23]. Thus, the socio-cultural adaptation of CMATMs not only relates to the psychological states and life quality of each member of the medical aid team, but also to the health aid performance of every single member and the medical aid team as a whole.

CMATMs may encounter multiple challenges in African countries, such as experiencing the differences between China and African countries in terms of the natural environments, social and cultural norms, and levels of social and economic development [24]. In addition, Chinese medical aid teams are often dispatched to regions of African countries that are facing hardships, which may bring greater difficulties and challenges. Studies have shown that, in these regions, water shortages, power outages, a lack of living materials, houses being in disrepair and old household appliances are the norm, and lagging network communication facilities make communication with relatives and friends inconvenient [24,25,26]. All of these changes and differences are stressors for CMATMs and they will inevitably affect their social-cultural adaptation, consequently influencing their feelings and opinions. Later, CMATM candidates will refer to the feelings and opinions of former CMATMs, hence impacting the willingness of later candidates to participate in similar programmes, which offers a possible explanation for the present low willingness to participate in medical aid programmes in African countries [27].

However, little has been determined so far regarding socio-cultural adaptation and its related factors for CMATMs. Therefore, the present study analyzed socio-cultural adaptation and its related factors for CMATMs in Africa and then provided scientific evidence for implementing relevant measures to improve the socio-cultural adaptation of this particular population.

## 2. Materials and Methods

### 2.1. Research Design and Procedure

This study used a cross-sectional questionnaire survey. Prior to conducting the data collection, we obtained permission and assistance from the related authorities and partner institutions, including the Department of International Cooperation and Development of the former National Health and Family Planning Commission of the People’s Republic of China, Peking University, and Shandong University. The relevant information of this study was effectively publicized in advance to all of the CMATM team leaders in Africa by the Department of International Cooperation and Development of the former National Health and Family Planning Commission of the People’s Republic of China while using a QQ (a mobile instant messaging app with the largest user base in China.) group chat.

We distributed our electronic questionnaires non-randomly to all of the CMATMs who were carrying out medical assistance missions in African countries during the period from 15 August 2016 to 20 November 2016 while using the information platform of the Global Health Institute of Wuhan University. We also asked respondents to answer all of the questionnaires independently and anonymously. No incentive was given to potential participants to participate. The finished questionnaires were checked to ensure the effectiveness of the questionnaires, and the incomplete or otherwise problematic questionnaires were not used.

### 2.2. Survey Tools

The questionnaire that was employed for this survey study consisted of the following two parts. The first part determined the demographic characteristics of the participants. Besides the common characteristics (sex, age, marital status, age, educational level, previous overseas experience, length of service), which are often adopted in adaptation researches [12,13,14,15,16,17], we included two particular characteristics in this study: type of service region and gross annual aid income.

CMATMs are dispatched to African countries with different natural environment, economic and social development levels, terrorist attack possibilities, etc. These differences bring different hardships, challenges, and difficulties for CMATMs. For example, CMATMs serving in the capital have better overall conditions than those serving in remote, undeveloped urban areas; those serving in areas that are often in shortage of water and electricity undoubtedly face greater difficulties than those serving in cities that have better public utilities and services. Hence, the Ministry of Finance of the People’s Republic of China classifies African host countries into different hardship regions according to their hierarchical hardships, and grants foreign aid allowances on this basis. Having consulted experts, government officers, and returned CMATMs, we argue that the type of service region (the hardship regions categorized by the Ministry of Finance of the People’s Republic of China) may impact the socio-cultural adaptation of our targeted population. Annual aid income was also taken into consideration, because it is obviously an important resource for CMTAMs to solve and alleviate unfavorable residential factors, hence it may consequently affect the socio-cultural adaptation of CMTAMs. Additionally, these two factors are in line with one of the aims of our project-offer suggestions for what Chinese government can do to enhance the socio-cultural adaption of CMATMs. The length of residence in host country has been proven to be an important factor [10]. In this study, we fine-tuned it to the length of service. Since recent studies have different findings and opinions on the length of service of CMATMs [28], it did make sense to exam the relationship between length of service and socio-cultural adaptation of CMATMs, and the results could provide reference for policy-makers.

Hence, we asked the participants’ sex (male/female), marital status (single/married), age group (≤40, 41~50, or 51~60), education level (junior college or below, bachelor’s degree, and master’s degree or above), previous overseas experience (yes/no), type of service region (non-hardship region, Class I hardship region, Class II hardship region, Class III hardship region, or Class IV hardship region, the last of which indicates the toughest living conditions), gross annual aid income (≤130,000 CNY, 130,000~190,000 CNY, 190,000~250,000 CNY, or >250,000 CNY), and length of service (≤6 months, 6–12 months, 12–18 months, or >18 months).

The second part adopted the Chinese version of the Socio Cultural Adaptation Scale (CSCAS) that was developed by Qi and Li [18]. The CSCAS has been widely used in many studies on socio-cultural adaptation among international students in China. For the targeted participants of this study, we fine-tuned a few items of the original scale (e.g., the term “campus” was replaced by the term “host country”.). The CSCAS contains 20 questions, and all of the questions use a five-point Likert scale, which ranged from very poor cross-cultural social adaptation (1 point) to very good adaptation (5 points). The CSCAS includes the dimensions of living environment, social morality, social service, social support, and interaction. The Cronbach’s alpha coefficient of the CSCAS is 0.93 in the present study.

### 2.3. Statistical Analysis

The Statistical Package for the Social Sciences (SPSS) version 22.0 for Windows (SPSS Inc., Chicago, IL, USA) was used to run all statistical analysis work, and the two-sided *p*-value of less than 5% for all the tests indicated statistically significant differences.

Data analysis was performed in three steps. First, the demographic characteristics and the scores of all dimensions of socio-cultural adaptation of the participants were both summarized via an initial descriptive analysis. The frequencies with proportions and mean values with standard deviations are presented in Table 1 and Table 2. Subsequently, we used t/F tests to compare the differences for the scores of all dimensions of socio-cultural adaptation between different population subgroups. In the final step, stepwise multiple linear regression models were performed to assess the potential related factors that might affect CMATMs’ socio-cultural adaptation in all dimensions.

### 2.4. Ethical Statements

Ethical approval for this study was obtained from the institutional review board of School of Health Science of Wuhan University (Project Identification Code: 2016-S-0011-4/03). The participants were guaranteed no risk being involved in participating in the survey, and the survey was conducted anonymously online.

## 3. Results

### 3.1. Demographic Characteristics of the Study Sample

Of the 317 respondents, the majority were male (65.62%). Most of the participants were married (85.80%) and they had no previous overseas experience (72.87%). Slightly less than half of the participants were in their forties (46.37%), and more than half (63.09%) had received an undergraduate education. Only 19.6% of the participants worked in non-hardship regions, and 32.18% of the participants earned between 130,000 and 190,000 CNY as their annual aid income. In terms of length of service, a service period of 12–18 months ranked at the top (39.12%), followed by a service period of over 18 months (28.08%). Table 1 summarizes the characteristics of the 317 participants.

### 3.2. Socio-Cultural Adaptation

Socio-cultural adaptation was measured while using five dimensions, including living environment, social morality, social service, social support, and interaction. The overall socio-cultural adaptation scores of the participants were 3.18 ± 0.62. The scores of social morality, living environment, social support, social service, and interaction were 3.38 ± 0.79, 3.26 ± 0.85, 3.18 ± 0.95, 3.08 ± 0.83, and 3.04 ± 0.73, respectively. The highest scores were observed in social morality, while the lowest scores were found in interaction (Table 2).

### 3.3. Factors

The overall adaption, social morality, and social service scores exhibited significant differences across the demographic characteristics, including type of service region, gross annual aid income, and length of service (Table 3).

In terms of social support, significant differences were found across demographic characteristics, including gross annual aid income, length of service, and age (Table 3). In the dimension of living environment, significant differences were detected across the type of service region and gross annual aid income. Table 3 also reveals significant differences across gross annual aid income and length of service in terms of interaction.

As demonstrated in Table 4, stepwise multiple linear regression models showed that the type of service region and gross annual aid income were related factors for overall socio-cultural adaptation and social morality. The groups of all kinds of hardship regions showed worse overall adaptation (all B < 0, *p* < 0.05) and social morality (all B < 0, *p* < 0.05) than the group of non-hardship regions did. When compared to non-hardship regions, those who worked in Class I (B = −0.49, 95% CI = −0.85, −0.13, *p* < 0.01), Class II (B = −0.56, 95% CI = −0.90, −0.22, *p* < 0.00), and Class III (B = −0.44, 95% CI = −0.82, −0.07, *p* < 0.05) showed worse adaptation in terms of social support. Regarding interaction, the groups of Class II (B = −0.28, 95% CI = −0.55, −0.01, *p* < 0.05) and Class IV (B = −0.35, 95% CI = −0.65, −0.05, *p* < 0.05) were worse than the group of the non-hardship region. In terms of living environment, those who served in Class I (B = −0.35, 95% CI = −0.64, −0.06, *p* < 0.05) and Class II (B = −0.33, 95% CI = −0.60, −0.06, *p* < 0.05) had worse adaptation than those who served in non-hardship region. Regarding social service, the group of Class I (B = −0.44, 95% CI = −0.76, −0.11, *p* < 0.01) showed worse adaptation than the group of non-hardship region did.

The group with higher annual aid income showed better overall adaptation (all B > 0, *p* < 0.05), social morality (all B > 0, *p* < 0.05), and social service (all B > 0, *p* < 0.05) than the group with ≤ 130,000 CNY did. Those who earned 190,000–250,000 CNY (B = 0.53, 95% CI = 0.21, 0.85, *p* < 0.001) and > 250,000 CNY (B = 0.52, 95% CI = 0.17, 0.88, *p* < 0.01) as their gross annual aid income had better social support adaptation than those who earned ≤ 130,000 CNY. Compared with the group with ≤ 130,000 CNY, the group with 190,000–250,000 CNY (B = 0.30, 95% CI = 0.04, 0.55, *p* < 0.05) showed better interaction, while the group with 130,000-190,000 CNY (B = 0.24, 95% CI = 0.01, 0.48, *p* < 0.05) showed better adaptation in terms of living environment.

In terms of length of service, those with a service period of over 18 months had better overall adaptation (B = 0.24, 95% CI = 0.03, 0.44, *p* < 0.05), social morality (B = 0.26, 95% CI = 0.01, 0.52 *p* < 0.05), social support (B = 0.58, 95% CI=0.28, 0.88, *p* < 0.05), and interaction (B = 0.33, 95% CI = 0.09, 0.57, *p* < 0.01) than those with a service period of ≤ 6 months.

## 4. Discussion

Dispatching medical aid teams is one of the most important ways in which China provides medical aid to African countries, and China has promised to continue this program. CMATMs will face pressure due to the great differences between China and African countries, including weather, diet, religions, and culture [29]. Moreover, CMATMs still have to face high risks in the form of infectious diseases and terrorist attacks. The destinations of Chinese medical aid teams usually have a high prevalence of infectious diseases, such as HIV, malaria, dengue fever, and Ebola. The Aid Worker Security Report 2018 reported that, in 2017, 139 aid workers were killed, 102 were wounded, and 72 were kidnapped [30]. As compared to 2016, 2017 saw a 30% rise in global fatalities. CMATMs’ adaptability to these challenges is bound to affect their socio-cultural adaptation. Therefore, this study aimed to reveal socio-cultural adaptation and its related factors for this population.

The hierarchical hardships in the overall environment and conditions of different areas indicates hierarchical changes, challenges, and possibilities for stresses among these areas, which might bring hierarchical difficulties and a failure to socio-culturally adapt. The Ministry of Finance of the People’s Republic of China classifies these hardship regions according to hierarchical hardships and grants foreign aid allowances on this basis. However, hierarchical hardship among the regions has seldom received attention in related research.

This study is among the first to explore the socio-cultural adaptation and its related factors for CMATMS in Africa. The CMATMs showed moderate overall socio-cultural adaptation despite the challenges, difficulties, and risks that are brought by undeveloped and hardship regions. This finding might be related to their profession—the education and work background of medical staff equip them with knowledge for coping with stress and adverse external environments.

Of the five dimensions, interaction scored the lowest, regardless of the degree of adaptation, which was similar to the results of previous studies [31]. The main reasons for this finding might be related to language barriers and cultural interaction differences. Liang focused on CMATMs that were sent by Guangxi Province, and he found that the participants considered the language barrier to be the second highest difficulty [25]. Wen’s study also demonstrated similar findings. In her study on 69 CMATMs sent by Jilin Province, nearly 50% of the participants indicated that the language barrier was their main difficulty [32]. Studies on similar populations have found similar results and explanations. Bjerneld et al. found that some of the Swedish medical personnel who had participated in health assistance programs felt isolated because of language barriers [33]. According to Chen’s study, voluntary Chinese language teachers in Thailand ranked language barriers and complex interaction relationships as the second and third main difficulties [34]. Another study found that the expatriates’ proficiency in the host country’s language was positively correlated with overall socio-cultural adaptation, especially in the dimension of interaction [35]. Regarding cultural interaction differences, a study on international higher degree research students in Australia who came from five different countries revealed that integration into the community, interaction with other students, and relationships with supervisors were a source of anxiety, frustration, and stress among the participants [36]. Relational skills are crucial for assignees, according to one article using data from 66 articles that involved 8474 (independent) overseas assignees [19]. Therefore, our study suggests that language level should be stressed in the selection criteria; meanwhile, language skills and cultural interaction skills should be highlighted in the training programme.

Our study found that the type of service region is associated with the overall socio-cultural adaptation. The overall socio-cultural adaptation was poorer than those of non-hardship regions. It is obvious that non-hardship areas enjoy better natural and social resources than hardship regions do, thus providing a better overall environment and bringing fewer challenges. The worse the hardship in the region is, the more difficulties and pressure the CMATMs may encounter. Therefore, priority should be given to improving the living conditions in hardship areas. In addition, we do not think that this finding is unique to CMATMs and we suggest that future studies investigate the role of hardship areas for other populations.

Annual aid income influenced overall socio-cultural adaptation, social morality, and social service. Our findings were in line with the findings of Huang. Huang’s results showed that higher income brought better cross-cultural adaptation of Chinese national expatriates [35]. It is understandable that annual aid income is an important resource for CMTAMs to solve and alleviate unfavorable residential factors. In addition, higher income allows for more types of leisure activities and more frequent leisure activities. According to Froese et al., leisure could be an effective facilitator of socio-cultural adaptation [31]. Hence, measures should be taken to increase the annual aid income of CMTAMs.

In line with prior studies [18,19,37,38], our findings demonstrated that the length of service was the related factor for overall adaptation. Service periods of over 18 months demonstrated the best overall adaptation in this study. A study found that the short-term medical expert group was more effective than the long-term medical aid team [28]. The “Brightness Action” Program targeting cataracts provides short-term medical aid. In this program, medical experts generally stayed at the destination for approximately one month. Since 2003, China has enacted this program in North Korea, Cambodia, Bangladesh, Vietnam, Pakistan, and other Asian countries [1]. Undoubtedly, a short-term program is effective in reducing the troubles and stresses of the overall changes for the medical staff and in treating certain diseases—for instance, cataracts and Ebola. However, such programs are not conducive to the follow-up treatment of some diseases. We argue that long-term and short-term medical aid cannot replace each other. Furthermore, with the developments in technology, telemedicine will bring opportunities for changing the medical aid model.

Previous research has demonstrated contradictory findings regarding the correlation between previous overseas experience and socio-cultural adaptation. McCall and Hollenbeck interviewed 101 managers who practiced a global career with world-renowned transnational enterprises. Their study found that the life or work experience gained from over half a year’s cross-cultural experience can be theorized in an individual’s subconsciousness and then unconsciously applied to a new cross-cultural environment, which makes the following or new adaptation easier [39]. However, Che found that previous overseas experience did not result in better adaptation [40]. The findings of our study were in line with those of Che’s study. First, the previous overseas experience of CMATMs might have been gained through holiday travel. Hence, the trips were often very short; moreover, the destinations were probably not hardship areas, as these were holiday trips. This kind of previous experience likely does not apply to the medical aid experience to Africa and may instead result in unrealistic expectations. Excessive expectations are not conducive to cross-cultural adaptation [31,41]. We suggest that attention be paid to reducing CMATMs’ expectations for the life of an aid worker, especially those CMATMs with previous overseas experience.

Last, a number of limitations of this study should be elucidated. First, this study only studied the socio-cultural adaptation of CMATMs and it did not involve psychological adaptation. Therefore, a comprehensive understanding of the cross-cultural adaptations of this population needs further research. Second, this study used a cross-sectional design. The correlation between variables and socio-cultural adaptation cannot be explained as causality, but it can only be seen as forming in a time segment in the process of multi-factor interaction. It is recommended that future studies be able to combine horizontal and vertical analysis. A longitudinal study is necessary to explore this issue further. Third, socio-cultural adaptation in a new cultural place should be from both sides-the side of the host country and the side of the sending state. The characteristics of the host country, the host country’s policy towards foreigners, and the social and cultural tolerance of the host country impact the overall socio-cultural adaptation regarding the side of the host country [42,43,44]. However, this study mainly explored “China’s side”, because: (1) the purpose of the project was to provide suggestions for China to improve CMATM policy, hence the research should focus on what China can do; (2) the medical aid teams are dispatched to about 30 African countries, hence it was impossible to explore the cultures of so many African countries in one paper. Further research could explore “Africa’s side” to better the understanding of the socio-cultural adaptation of CMATMs in particular African countries. Moreover, 18 poorly completed questionnaires (e.g., the answers were exactly the same for all questions and/or more than a third of the questions were not answered) were excluded, and there might be more information on this, yet we did not track in the present study. It is worth noting that this is the first study in China that aims to explore the current situations of socio-cultural adaptation and its related factors for CMATMs in Africa. Additionally, the characteristics for the research population so far are not available. For this reason, it currently remains unclear whether the sample from the present survey is compositionally similar to the research population. Hence, further studies are needed for more in-depth explorations on these aspects.

## 5. Conclusions

This study is among the first to explore the socio-cultural adaptation and its related factors for CMATMSs in Africa. The results revealed that the overall socio-cultural adaptation of respondents was moderate. The hardship classification of the areas was the related factor of overall socio-cultural adaptation and every dimension. Groups with higher annual aid income showed better overall adaptation, social morality, and social service than the group receiving < 13 million yuan. In terms of service period, those with a service period of over 18 months had better overall adaptation, social morality, social support, and interaction than did those with a service period of less than six months. The present study indicated the possibility that improving living conditions in hardship areas, increasing the annual aid income, and reinforcing the language, interaction culture and skills could be an effective way to promote socio-cultural adaptation for CMATMs in Africa.

## Figures and Tables

**Table 1 ijerph-16-03132-t001:** Characteristics of the study population (n = 317).

Characteristics	n	%
**Sex**		
Female	109	34.4
Male	208	65.6
**Age**		
≤40	98	30.9
41~50	147	46.4
51~60	72	22.7
**Marital status**		
Single	45	14.2
Married	272	85.8
**Education level**		
Junior college or below	34	10.7
Bachelor’s degree	200	63.1
Master’s degree or above	83	26.2
**Overseas experience**		
No	231	72.9
Yes	86	27.1
**Type of service regions**		
Non-hardship region	62	19.6
Class I hardship region	79	24.9
Class II hardship region	87	27.4
Class III hardship region	44	13.9
Class IV hardship region	45	14.2
**Gross annual aid income**		
≤130,000	61	19.2
130,000~190,000	102	32.2
190,000~250,000	97	30.6
>250,000	57	18.0
**Length of service**		
≤6 months	70	22.1
6~12 months	34	10.7
12~18 months	124	39.1
>18 months	89	28.1
**Overall**	317	100.0

**Table 2 ijerph-16-03132-t002:** Socio-cultural adaptation scores of the study population (data presented as mean ± standard deviation).

Dimensions	mean ± standard Deviation
Social morality	3.38 ± 0.79
Living environment	3.26 ± 0.72
Social support	3.18 ± 0.95
Social service	3.08 ± 0.83
Interaction	3.04 ± 0.73
Overall socio-cultural adaptation	3.18 ± 0.62

**Table 3 ijerph-16-03132-t003:** Single related factors of overall socio-cultural adaptation and adaptation of each dimension.

Variables	Social Support	Social Morality	Social Service	Interaction	Living Environment	Overall
Sex	t = 1.04	t = 0.93	t = 1.13	t = 1.48	t = −0.09	t = 1.06
Female	3.26 ± 0.90	3.43 ± 0.82	3.16 ± 0.75	3.12 ± 0.67	3.25 ± 0.68	3.23 ± 0.59
Male	3.14 ± 0.97	3.35 ± 0.78	3.04 ± 0.87	3.00 ± 0.76	3.26 ± 0.74	3.15 ± 0.64
Age	F = 3.21 *	F = 1.01	F = 0.27	F = 0.16	F = 0.11	F = 0.20
≤40	2.99 ± 1.04	3.46 ± 0.81	3.03 ± 0.92	3.01 ± 0.74	3.26 ± 0.68	3.16 ± 0.63
41~50	3.23 ± 0.92	3.32 ± 0.77	3.11 ± 0.79	3.05 ± 0.74	3.24 ± 0.68	3.17 ± 0.61
51~60	3.35 ± 0.87	3.40 ± 0.82	3.09 ± 0.80	3.07 ± 0.72	3.28 ± 0.85	3.22 ± 0.66
**Marital status**	t = −0.05	t = −0.49	t = −0.40	t = −1.43	t = −0.70	t = −0.93
Single	3.18 ± 0.99	3.33 ± 0.86	3.04 ± 0.84	2.90 ± 0.76	3.19 ± 0.64	3.10 ± 0.63
Married	3.19 ± 0.95	3.39 ± 0.78	3.09 ± 0.83	3.07 ± 0.73	3.27 ± 0.73	3.19 ± 0.62
**Education level**	F = 2.74	F = 1.80	F = 1.43	F = 0.38	F = 0.60	F = 1.35
Junior college or below	3.50 ± 0.89	3.62 ± 0.87	3.27 ± 0.85	3.15 ± 0.88	3.38 ± 0.80	3.34 ± 0.70
Bachelor’s degree	3.19 ± 0.93	3.34 ± 0.79	3.09 ± 0.81	3.03 ± 0.72	3.25 ± 0.73	3.17 ± 0.61
Master’s degree or above	3.05 ± 1.02	3.38 ± 0.75	3.98 ± 0.86	3.04 ± 0.71	3.22 ± 0.66	3.14 ± 0.61
**Overseas experience**	t = −1.31	t = −0.75	t = −0.59	t = −0.77	t = 1.80	t = 2.46
No	3.14 ± 0.90	3.36 ± 0.75	3.07 ± 0.80	3.02 ± 0.69	3.29 ± 0.75	3.17 ± 0.61
Yes	3.31 ± 1.07	3.43 ± 0.89	3.13 ± 0.91	3.09 ± 0.84	3.15 ± 0.63	3.19 ± 0.67
**Type of service regions**	F = 2.16	F = 2.64 *	F = 2.73 *	F = 1.15	F = 2.586 *	F = 2.46 *
Non-hardship region	3.39 ± 0.78	3.64 ± 0.73	3.29 ± 0.72	3.21 ± 0.60	3.48 ± 0.69	3.38 ± 0.55
Class I hardship region	3.06 ± 1.11	3.35 ± 0.79	2.87 ± 0.96	3.01 ± 0.86	3.13 ± 0.67	3.21 ± 0.55
Class II hardship region	3.07 ± 1.02	3.33 ± 0.84	3.10 ± 0.88	2.98 ± 0.79	3.17 ± 0.77	3.12 ± 0.71
Class III hardship region	3.11 ± 0.89	3.17 ± 0.73	3.08 ± 0.63	3.09 ± 0.59	3.25 ± 0.62	3.15 ± 0.50
Class IV hardship region	3.42 ± 0.71	3.38 ± 0.78	3.15 ± 0.75	3.96 ± 0.65	3.35 ± 0.80	3.08 ± 0.65
**Gross annual aid income**	F = 6.93 **	F = 3.28 *	F = 6.81 **	F = 3.33 *	F = 2.58 *	F = 5.67 **
≤130,000	2.71 ± 1.06	3.10 ± 0.73	2.67 ± 0.87	2.79 ± 0.69	3.05 ± 0.59	2.89 ± 0.61
130,000~190,000	3.22 ± 0.96	3.48 ± 0.80	3.16 ± 0.83	3.06 ± 0.73	3.37 ± 0.80	3.25 ± 0.64
190,000~250,000	3.36 ± 0.89	3.44 ± 0.79	3.17 ± 0.76	3.16 ± 0.77	3.27 ± 0.68	3.26 ± 0.59
>250,000	3.34 ± 0.77	3.39 ± 0.81	3.24 ± 0.79	3.10 ± 0.67	3.25 ± 0.73	3.23 ± 0.59
**Length of service**	F = 6.22 **	F = 2.81 *	F = 3.23 *	F = 4.66 *	F = 1.34	F = 4.22 *
≤6 months	2.78 ± 1.04	3.16 ± 0.69	2.82 ± 0.90	2.76 ± 0.66	3.14 ± 0.62	2.95 ± 0.58
6~12 months	3.24 ± 0.80	3.49 ± 0.84	3.16 ± 0.78	3.19 ± 0.76	3.23 ± 0.82	3.25 ± 0.73
12~18 months	3.25 ± 0.81	3.38 ± 0.70	3.20 ± 0.70	3.10 ± 0.62	3.35 ± 0.67	3.25 ± 0.53
>18 months	3.40 ± 1.04	3.51 ± 0.93	3.10 ± 0.93	3.13 ± 0.86	3.23 ± 0.81	3.24 ± 0.70
**Overall**	3.18 ± 0.95	3.38 ± 0.79	3.08 ± 0.83	3.04 ± 0.73	3.26 ± 0.72	3.18 ± 0.62

Note: “*” *p*-value < 0.05; “**” *p*-value < 0.01.

**Table 4 ijerph-16-03132-t004:** Linear regression models of related factors for overall socio-cultural adaptation and adaptation of each dimension.

Variables	Social Support		Social Morality		Social Service		Interaction		Living Environment		Overall	
	*B*	(95% *CI*)	*B*	(95% *CI*)	*B*	(95% *CI*)	*B*	(95% *CI*)	*B*	(95% *CI*)	*B*	(95% *CI*)
**Type of service regions**	-	-	-	-	-	-	-	-	-	-	-	-
Non-hardship region			-	-	-	-	-	-	-	-	-	-
Class I hardship region	−0.49	(−0.85, −0.13) **	−0.41	(−0.72, −0.10) **	−0.44	(−0.76, −0.11) **	−0.22	(−0.51,0.07)	−0.35	(−0.64, −0.06) *	−0.34	(−0.59, −0.10) **
Class II hardship region	−0.56	(−0.90, −0.22) ***	−0.44	(−0.73, −0.14) **	−0.22	(−0.53, 0.09)	−0.28	(−0.55, −0.01) *	−0.33	(−0.60, −0.06) *	−0.34	(−0.57, −0.11) **
Class III hardship region	−0.44	(−0.82, −0.07) *	−0.53	(−0.85, 0.21) ***	−0.28	(−0.62, 0.07)	−0.16	(−0.46, 0.14)	−0.24	(−0.54, 0.07)	−0.28	(−0.54, −0.03) *
Class IV hardship region	−0.25	(−0.63, 0.13)	−0.40	(−0.72, −0.07) *	−0.25	(−0.58, 0.09)	−0.35	(−0.65, −0.05) *	−0.15	(−0.45, 0.15)	−0.27	(−0.52, −0.02) *
**Gross annual aid income**	-	-	-	-	-	-	-	-	-	-	-	-
≤130,000			-	-	-	-	-	-	-	-	-	-
130,000–190,000	0.28	(−0.02, 0.58)	0.29	(0.03, 0.55) *	0.41	(0.13, 0.68) **	0.16	(−0.08, 0.41)	0.24	(0.01, 0.48) *	0.25	(0.05, 0.46) *
190,000–250,000	0.53	(0.21, 0.85) ***	0.38	(0.11, 0.65) **	0.49	(0.21, 0.77) ***	0.30	(0.04, 0.55) *	0.23	(−0.02, 0.48)	0.34	(0.13, 0.55) **
>250,000	0.52	(0.17, 0.88) **	0.36	(0.06, 0.67) *	0.57	(0.25, 0.89) ***	0.26	(−0.02, 0.55)	0.18	(−0.10, 0.46)	0.33	(0.09, 0.56) **
**Length of service**	-	-	-	-	-	-	-	-	-	-	-	-
≤6 months			-	-	-	-	-	-	-	-	-	-
6–12 months	0.12	(−0.29, 0.53)	0.11	(−0.24, 0.46)	-	-	0.29	(−0.03, 0.62)	-	-	0.11	(−0.17, 0.38)
12–18 months	0.18	(−0.13, 0.48)	0.01	(−0.25, 0.26)	-	-	0.19	(−0.05, 0.43)	-	-	0.10	(−0.09, 0.31)
>18 months	0.58	(0.28, 0.88) ***	0.26	(0.01, 0.52) *	-	-	0.33	(0.09, 0.57) **	-	-	0.24	(0.03, 0.44) *

Note: B = Coefficient; CI = Confidence Interval; “*” *p*-value < 0.05, “**” *p*-value < 0.01, “***” p-value < 0.001.

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
