# Peer review of "Socio-Cultural Adaptation and Its Related Factors for Chinese Medical Aid Team Members (CMATMs) in Africa"

_ijerph, 2019, doi:10.3390/ijerph16173132_

Round 1

Reviewer 1 Report

This is a solid manuscript—well-written, practical, and clearly within the field of research of cross-cultural adaptation.

I have only a couple of substantive suggestions, and these involve a clearer framing of the work within Colleen Ward’s larger approach to cross-cultural adaptation (see notes below), clearer definition of terms within the sociocultural adjustment scale, and some clarifications within the results section.
25-32: The abstract is very solid. It might have more detail than it needs (e.g., you could delete specific statistics and just keep the conclusions from the findings in the abstract).
56: The general line of the argument as a whole is clear. One of my main recommendations is to give more context—maybe a paragraph—that describes Colleen’s approach to adjustment (that is, that includes both psychological and sociocultural adjustment). I think it would be good to cite her by name, discussing these dimensions—or domains--of adjustment (I think Ward has written with John Berry as well as alone to clarify these two dimensions). This will also give context to the limitation that the study does not include “psychological adaptation.” Part of the more developed discussion (based on some of her research) can explain how the two notions are distinct but related (this is important later in your argument when you link sociocultural adjustment with depression, since depression is really a key aspect (and usual measurement) of psychological adjustment. If it is available to you, you could also frame this theoretical paragraph in terms of Jefferey Ady’s (1995) chapter (Wiseman’s edited book on intercultural communication theory) that argues for a “differential demand” approach to adjustment. Ady also argues that adjustment occurs in different domains.
The argument above *might* suggest moving the “Furthermore” paragraph (lines 72ff) to above “Studies” (lines 57 ff). That is, the line of argument here might be: 1) domains of adjustment and how sociocultural adjustment fits in here; 2) general findings related to sociocultural adjustment, to demonstrate why it merits specific focus; 3) how/why sociocultural adjustment relates specifically to CMATMs. [Then, later, when you mention depression, refer back to psychological adjustment).
My second main suggestion is to provide clearer definition of some terms. In some cases, these are terms or phrases that occur as you pass through the argument (what, for example, is “sociocultural adaptation”? what does “expatriate willingness” mean? (This is important because it seems that sociocultural adjustment is a measurement of adjustment of someone on the field, but, depending on what you mean by “expatriate willingness,” this would be something experienced by someone who has not yet traveled yet. You do make an argument for this connection, but it is as not as strong as other reasons you present for studying sociocultural adaptation. What are “psychological withdrawal cognitions?” Most importantly, what are the dimensions of sociocultural adjustment (since you measure these and discuss them in your findings). These might warrant another short paragraph after you introduce the domains.
My own preference is for more overt RQs, but you do have a clear statement of purpose for the study (80-83), so this is good enough for me.
METHOD: The method of the study seems solid, rigorous. Clarify the total number of surveys sent and the response rate. Was any incentive given to potential participants to participate?
My own preference (and that of the American Psychological Association, I believe) is to clearly differentiate between sex and gender. I suspect, if you merely asked if participants were male or female, that they responded in terms of their biological make-up (sex), and not considering their enactment of what it means to be male or female (gender).
Very clear on data collection and analysis. It looks like you used F tests (was it ANOVA?) to determine if differences existed within variables with multiple categories in the predictor variable (e.g., level of hardship), using t-tests for follow up pair-by-pair comparisons. If this is so, make this point clearer. Normally in such cases, I am used to researchers using something like Scheffé or some other “post hoc” analysis that SPSS allows as options in ANOVA analysis.
RESULTS: The results are mostly well-reported, though I have a couple of questions. Table 4 shows stepwise multiple regression. Did you the variables in any “blocks” (e.g., first entering demographic variables?). If so, it would be good for the reader to know that.
172: “those worked” à “those who worked”—this is one of the only writing errors I found in the whole document! (very nice)
Consider noting up front that there were no differences in sociocultural adjustment based on sex, marital status, or educational level (at least if I am understanding the stats correctly.
On table 3 (and any other table that goes to 2 pp), put column headings on top of 2nd Add a “note” at the bottom of table, either to note that * = p < .05 OR . . . see next note.
I found some of the findings confusing (or at least incomplete). Just as an example, the results section notes that those with higher income core higher on overall adaptation, social morality, and social service; but the findings suggest that they were higher in all dimensions (also, since at the end of the stats, you note that “all B > 0, p < .05,” you don’t need that after each specific score). Even this conclusion, however, may not be accurate to the table. For example, you note higher scores for those with the highest income on overall adaptation (but 190-250K has a higher mean), social morality (though two of the income brackets have higher means than those with the highest incomes), and social service (the only item where highest income bracket, indeed, has highest mean). What I recommend here is twofold and includes both the table and the write up. First, on the table consider marking within the table means *within* each category that show significant differences. Recalling that a significant “F” only means that there is significance somewhere in the results, your follow-up t-tests (probably a Scheffé or some other text, rather than a t-test) could show exactly where the difference(s) lie(s). I don’t have the stats to know, but, just for example, under “Gross annual aid income” you could do this (though subscripted letters may be lost through review software formatting). And 2) once which specific differences are significant is clear, make sure that the summary of findings in paragraph lines 170-182 aligns precisely with the findings. But, as reported, findings seem clear, and I don’t think you need more detail in actual reporting of findings.

Gross annual aid income       F=6.93*

≤130,000                                 2.71±1.06abc

130,000-190,000                      3.22±0.96ad

190,000-250,000                      3.36±0.89bd

≥ 250,000                                3.34±0.77c

______

*p < .05; similar subscripts within a set of means indicates significant differences, p < .05

Table 4 is not in the format that I usually expect to see multiple regression reports in. Rather than a table with each of the categories of a variable (e.g., hardship) listed, as if entered separately, I would expect the variables (e.g., hardship, gross annual aid, etc.) entered as separate predictor variables, more as how appear in these examples: https://www.ncfr.org/sites/default/files/2017-01/regression.pdf. At the same time, such a table would be helpful, but I don’t know that it would be essential for the manuscript; I would yield to the other reviewer on this point.
Probably move multiple regression paragraph to after discussion of Table 3 findings. The things noted in this paragraph, that such-and-such variables are “influencing factors” (if these stats allow us to determine causality, and not just relationship), though this doesn’t tell us much more than table 3 does (except that table 3 clearly shows only relationship, not causality. I don’t know about the assumptions of regression if it allows one to determine causal order). Would you be able, with the regression, to tell reader which variables contribute the most to sociocultural adaptation?
DISCUSSION: Overall, writing has same high quality as rest of the paper, with clear, strong integration with previous literature.
194: Format: Decide if you will use the Oxford comma with lists; follow whichever rule you choose consistently across the paper (e.g., compare with line 226)
The numbers indicated in the Aid Worker Security Report (aid workers killed, etc.)—are these worldwide, or only Chinese?
199: Wording: While the wording is interesting, I’m not sure what you mean by “hierarchical hardships,” “changes” and "difficulties”—what makes them “hierarchical”?
218: Format: Magdalena Bjerneld: Decide up front if you will include given names along with family names and follow whichever approach you decide to use consistently (if Magdalena is part of author’s family name, then ignore this comment).
211 ff: Great paragraph on language difficulties, with links to previous research. Word choice (227): probably “one article using data” rather than “study” including “studies”
240ff: After which differences are significant, make sure discussion lines up precisely with significant findings. At least if I use table 3, annual aid predicts all aspects measured (though the highest income individuals are not the most adapted in every area). Table 4 seems to suggest the same thing. I did not make the same comparison for all of discussion, i.e., to verify that each claim in the discussion is supported by the statistics.
Great practical recommendations, based clearly on findings, throughout
261: format: Probably McCall and Hollenbeck (not &); join “sub” and “conscious” together to form one word

Other than that, the MS looks good.

Author Response

Dear Editors and Reviewers,

The authors very much appreciate the thoughtful and critical feedback from the reviewers and editors. We are delighted at your decision that we revise and resubmit this manuscript.

The reviewers are clearly familiar with the topic, as is exemplified by their close and accurate review of this manuscript.

The manuscript has been revised according to the reviewers’ comments, and all changes have been highlighted for ready identification. In addition, we have addressed each of the comments from you specifically, and the following points are the replies to editor and each reviewer point by point.   

Hope the revision is satisfactory and this manuscript is now acceptable for publication in your journal.

* Please be noted that the page and line number may be changed after MS is changed.

I am looking forward to hearing from you soon.

Sincerely,

Authors

Reviewer 2 Report

This paper mainly talks about the the social-cultural adaptation of Chinese Medical Aid Team in Africa and stated a number of challenges. My suggestion is that while social-cultural adaptation in a new cultural place should be from both sides, it will be more comprehensives if examples of social-cultural practices from Africa can be included to support the arguments in the article. 

This article seems to have an one-sided view of the whole picture. While this is an article of social-cultural adaptation of Chinese in Africa, the paper should also explain situations in Africa (particularly examples in different African countries if possible) to show a more complete picture. Otherwise, the article will not be able to present a comprehensive picture of the topic.

Another suggestion is that Africa is continent with different countries of diverse social-cultural practices. It will be more accurate if the article can include examples from some African countries can be included. Otherwise, it may give the readers there will be a "shared" social-cultural practice in Africa.

Author Response

(The authors gave the same response as above.)

Reviewer 3 Report

This is an interesting paper that analyses socio-cultural adaptation of Chinese medical aid team members (CMATMs) in Africa using appropriate and unique survey data. The societal relevance of such research is evident, and the findings raise important questions for policy makers. However, I have a number of concerns about the theoretical framework, academic relevance, data source and empirical strategy.

Comment 1

The theoretical framework is currently too meagre. I am confused about the exact focus of the paper. While it is clear that socio-cultural adaptation plays a central role, it is not so clear to me how this is positioned in the wider literature. In lines 57-58, it is stated that socio-cultural adaptation has an effect on expatriate willingness. As I understand it, expatriate willingness is understood here as the desire to stay involved in this line of work abroad. In that context, socio-cultural adaptation would be the main independent variable. In this paper however, socio-cultural adaptation is the dependent variable. So if I understand correctly, the idea is to better understand determinants of socio-cultural adaptation, which is of interest in light of its association with expatriate willingness. In other words, the focus is not really on socio-cultural adaptation, but rather on its subsequent relevance for expatriate willingness. That to me raises the question why we would not just measure expatriate willingness directly. Either the focus on socio-cultural adaptation needs to be clarified, or the dependent variable needs to change.

More importantly, I find the predictors for socio-cultural adaptation highly arbitrary. The authors use a number of demographic characteristics, but there is no indication (1) why these were chosen, and (2) how they are expected to matter exactly. Exacerbating the confusion is line 110, stating that these indicators (the IVs) are the confounding factors in the study. Confounding what exactly? Are they not the determinants? Because of this ambiguity, it is unclear how the findings build on existing research. Is there no research on the determinants of socio-cultural adaptation? If not, then it is necessary for the authors to formulate a theoretical framework that details the mechanisms underlying the presumed relationship between the demographic characteristics and socio-cultural adaptation.

In sum, the choice of dependent and independent variables requires elaboration in my view.

Comment 2

To assess the quality of the findings, and interpret these effectively, more information on the data are necessary. In line 99-100, it is stated that finished questionnaires were checked for effectiveness, and that incomplete or problematic questionnaires were not used. The authors gloss over this important point too readily. What exactly constitutes a problematic questionnaire? How many missings, and on what items, were required before a questionnaire was no longer considered? How many were removed as a result? This is important, because non-respons is typically not random. Simply removing questionnaires on the basis of missings (or other problems, although it is unclear what that is) is thus likely to result in selection bias. This is especially problematic in light of the dependent variable. It seems highly plausible to me that socio-culturally adaptation is related to responding to the questionnaire.

I would like to see more information on non-response and would be interested to know what ‘problematic’ questionnaires are exactly. Moreover, I would like to see some reflection on the potential for selection bias, and how the findings should (or can) be interpreted in that light.

Comment 3

Related to the previous point, I am interested in the extent to which the data are representative of the population. While Table 1 is useful in this regard, I would like to see the same characteristics for the research population (in so far as they are available). Otherwise, there may be serious limits to the external validity of the findings.

Comment 4

The Chinese version of the Socio Cultural Adaptation Scale plays a crucial role in the operationalisation of the dependent variable. While I recognize that this scale is used widely in the literature, I would still like to see the specific components that make up the 5 dimensions. However, the only reference that is provided (in line 112) is in Chinese. I think it is crucial that the authors provide an English translation of the individual components that make up the 5 dimensions (possibly for an annex), so that the findings can be interpreted effectively.

Comment 5

I am confused about the regression analysis. It seems that most of the independent variables were not included in Table 4. Why is that the case? If they are confounders, then they per definition have to be included in the regression. I also think the authors should discuss the assumptions underlying linear regression analysis. For instance, are the dependent variables normally distributed? In other words, why is linear regression most appropriate there, as opposed to alternatives?

Author Response

(The authors gave the same response as above.)

Round 2

Reviewer 1 Report

The revision addresses my recommendations in the Rev of Lit, which is excellent. The overall quality of the MS is very solid in most respects, especially review of lit, method, and discussion. My main questions now still concern the results. In sum, almost all of my comments regard the clarity of the tables and the interpretation of these tables. It might be that you are drawing your conclusions mostly from Table 4 (which to be honest, I’m not sure I understand, as the scores there do not seem to compare items within a subgroup against each other).

There are maybe two or three writing errors in the whole document. Writing is mostly excellent. Here are my line-by-line comments:

71, 73: Probably “the” dimension in each case.

75-78: This general definition of sociocultural adaptation seems to repeat definition from above—or at least it seems to return to a general definition of sociocultural adaptation after you have already generally introduce dit and then moved to specific examples. Consider moving these lines to beginning of paragraph that starts line 69.

97: Delete “and” before “consequently”

215ff: Are the conclusions in this paragraph reliant on Table 4, Table 3, or a combination of these? As I understand Table 4, the item in each column indicates a level of relationship of predictability. For example, Class I hardship region predicts social support significantly (p < .01). It looks like the numbers “(-.085, -0.13)” are the CI (confidence interval?). But without more explanation, I’m not sure how to interpret that. Often, with multiple regression, the sort of question that I would expect is *which* of the variables (e.g., income, region hardship, length of service) is the *best* predictor of the variable in question.

Regardless, it seems on Table 4 that the stats are only the level and significance of predictability of each variable in the row as it predicts the variable in the column heading. That would not allow the type of comparison described on lines 215-227 (such-and-such is more than this-and-that).  So, the paragraph uses stats from Table 4, but seems to make the sort of more-or-less comparisons that only Table 3 allows (unless I’m not understanding the stats! I would yield to the other reader, as stats are not my strong point). Using both tables, I understood the stats fairly well, though I had some questions mostly in this paragraph and in the discussion.

215-217: all hardship regions worse overall adaptation and social morality (but the scores were also higher for “no hardship” regions on social service and living environment (as we would expect). Regarding those and other scores, as I noted on first draft, the significant F (e.g., F = 2.64* for Social Morality and Type of Service Region) only indicates that somewhere among the means under that section, there is some significant difference, but it does not tell us which of the different scores are significantly different. So, while 3.64 is higher than any of the other means in this section, we don’t know if it is *significantly* higher without post-hoc analyses of some sort (and indication in the table of *which* means are significantly different). *IF* you have run the post-hoc tests and are just nor reporting them, so that you know that, for the two items you mention on lines 215-216, “non-hardship” is *significantly* higher than each of the other means in the section, let the reader know that this is a claim based on such a comparison (maybe with footnote that indicates that follow-up scores are available on request?) 217: “Similar results were found for social support” –I’m not sure what this claim means, especially since, for social support, the mean for Class IV region was higher than the mean for the non-hardship region (though the means are close and likely not significantly different). 219-220: “Regarding interaction” [delete “the”]… On Table IV, Class IV interaction adjustment has a mean of 3.96, whereas non-hardship mean is 3.21. Again, Table 4 seems to indicate the direction and strength of each individual relationship and not how high adaptation was for any particular item. You have plenty of sources for the paper, but *if* your findings indeed support that social support and interaction are higher for areas with greater hardship, it might be that people in those regions recognize the hardship and are giving more support to the foreign aid workers, whereas, as difficulty decreases, the help and social support are withdrawn and the workers are left increasingly on their own. But this is just a passing thought. 221-224: Higher annual aid income better overall adaptation, social morality, and social service than 130,000CNY. The reporting hides some nuance in the stats on Table 3, e.g., that those with “middle” incomes scored higher than those with the highest income on overall adaptation, and the second lowest bracket (130-190) had a higher mean on social morality than the two highest brackets. The claim does hold true regarding social service. This could be a question of what you mean by “the groups with higher annual aid income”—do you mean the *highest* income, or simply any income higher than <130,000 CNY? If you mean the latter, you might say, rather than “higher annual income” something like “any income higher than the minimum” 224-227: Length of service: those with period over 18 month scored higher in this, that, and the other, but higher than which comparison group—those with less than 6 months, or all of the other groups? Tables: If possible, break Table 3 above “Type of Service Regions,” so that all of the subcategories appear on one page. Also, I think that you are supposed to repeat the column headings on top of the second page of a table. Good on p-value NOTE below table, though indication of specific significant means within each variable would be useful for drawing more meaningful conclusions. 257: related to language barriers [add “s”] 266: “host country’s language level”—I’m not sure what you mean, here. 275ff: Be careful of some claims here. For example, “Our study found that the type of service region influenced the overall socio-cultural adaptation and adaptation of every dimension”: But on Table 3, Class IV had a higher mean than non-hardship in social support and in interaction. 284: “Annual aid income influenced overall sociocultural adaptation, social morality, and social service.” Beyond the limitation that you note below, that your stats allow the measurement of correlation but not causality, based on F scores, you could say that annual aid influences (or rather, predicts) *every* aspect of sociocultural adaptation (all Fs are significant), though which means are highest differ for each dimension of sociocultural adjustment. Indeed, as I read Table 3, the highest level of national aid related (possibly) to the sub-area of social service. 292ff: Length of service: “Service periods of over 18 months demonstrated the best overall adaptation of the study”; but on Table 3, service periods of 6-12 and 12-18 months both had higher means than over 18 months (though means are close and probably not significantly different).

Note that if there are changes to the interpretation of the findings based on my notes here, that may lead to changes in the focus and content of some of the explanations in the discussion; however, if the claims remain unchanged, the discussion section does exactly what I would expect a discussion section to do.

306: McCall and Hollenbeck: Not in reference list (I did not compare all citations against the reference—I was just curious and wanted to look this one up and realized that there is no corresponding reference number)

308: subconscious (1 word)

325: “should be from both sides”—great point! But clarify the wording. If you mean that both sides need to adjust, this is a solid point to make.

Author Response

Review Report (Round 2)

Manuscript ID:

ijerph-563015

Title:

"Socio-cultural Adaptation and Its Related Factors for Chinese Medical Aid Team Members (CMATMs) in Africa"

Dear Editors and Reviewers,

We very much appreciate the thoughtful and critical feedback from the reviewers again. We are delighted at your decision that we revise and resubmit this manuscript. The reviewers have requested some further clarification and revisions to our manuscript, which we have completed, and all changes have been highlighted for ready identification. In addition, we have addressed each of the comments from you specifically, and the following points are the replies to editor and each reviewer point by point.

* Please note that the page and line number may be changed after MS is changed.

We are looking forward to hearing from you soon.

Sincerely,

Authors

Response to Reviewer 1 Comments

Point 1: 71, 73: Probably “the” dimension in each case.

Response 1: We have revised according to the suggestion.

Point 2: 75-78: This general definition of sociocultural adaptation seems to repeat definition from above—or at least it seems to return to a general definition of sociocultural adaptation after you have already generally introduce dit and then moved to specific examples. Consider moving these lines to beginning of paragraph that starts line 69.

Response 2: We have moved line 75-78 to beginning of paragraph that starts line 69.

Point 3:97: Delete “and” before “consequently”.

Response 3: We have deleted “and”.

Point 4: 215ff Are the conclusions in this paragraph reliant on Table 4, Table 3, or a combination of these? As I understand Table 4, the item in each column indicates a level of relationship of predictability. For example, Class I hardship region predicts social support significantly (p < .01). It looks like the numbers “(-.085, -0.13)” are the CI (confidence interval?). But without more explanation, I’m not sure how to interpret that. Often, with multiple regression, the sort of question that I would expect is *which* of the variables (e.g., income, region hardship, length of service) is the *best* predictor of the variable in question.

Regardless, it seems on Table 4 that the stats are only the level and significance of predictability of each variable in the row as it predicts the variable in the column heading. That would not allow the type of comparison described on lines 215-227 (such-and-such is more than this-and-that).  So, the paragraph uses stats from Table 4, but seems to make the sort of more-or-less comparisons that only Table 3 allows (unless I’m not understanding the stats! I would yield to the other reader, as stats are not my strong point). Using both tables, I understood the stats fairly well, though I had some questions mostly in this paragraph and in the discussion.

215-217: all hardship regions worse overall adaptation and social morality (but the scores were also higher for “no hardship” regions on social service and living environment (as we would expect). Regarding those and other scores, as I noted on first draft, the significant F (e.g., F = 2.64* for Social Morality and Type of Service Region) only indicates that somewhere among the means under that section, there is some significant difference, but it does not tell us which of the different scores are significantly different. So, while 3.64 is higher than any of the other means in this section, we don’t know if it is *significantly* higher without post-hoc analyses of some sort (and indication in the table of *which* means are significantly different). *IF* you have run the post-hoc tests and are just nor reporting them, so that you know that, for the two items you mention on lines 215-216, “non-hardship” is *significantly* higher than each of the other means in the section, let the reader know that this is a claim based on such a comparison (maybe with footnote that indicates that follow-up scores are available on request?) 217: “Similar results were found for social support” –I’m not sure what this claim means, especially since, for social support, the mean for Class IV region was higher than the mean for the non-hardship region (though the means are close and likely not significantly different). 219-220: “Regarding interaction” [delete “the”] On Table IV, Class IV interaction adjustment has a mean of 3.96, whereas non-hardship mean is 3.21. Again, Table 4 seems to indicate the direction and strength of each individual relationship and not how high adaptation was for any particular item. You have plenty of sources for the paper, but *if* your findings indeed support that social support and interaction are higher for areas with greater hardship, it might be that people in those regions recognize the hardship and are giving more support to the foreign aid workers, whereas, as difficulty decreases, the help and social support are withdrawn and the workers are left increasingly on their own. But this is just a passing thought. 221-224: Higher annual aid income better overall adaptation, social morality, and social service than 130,000CNY. The reporting hides some nuance in the stats on Table 3, e.g., that those with “middle” incomes scored higher than those with the highest income on overall adaptation, and the second lowest bracket (130-190) had a higher mean on social morality than the two highest brackets. The claim does hold true regarding social service. This could be a question of what you mean by “the groups with higher annual aid income”—do you mean the *highest* income, or simply any income higher than <130,000 CNY? If you mean the latter, you might say, rather than “higher annual income” something like “any income higher than the minimum” 224-227: Length of service: those with period over 18 month scored higher in this, that, and the other, but higher than which comparison group—those with less than 6 months, or all of the other groups? Tables: If possible, break Table 3 above “Type of Service Regions,” so that all of the subcategories appear on one page. Also, I think that you are supposed to repeat the column headings on top of the second page of a table. Good on p-value NOTE below table, though indication of specific significant means within each variable would be useful for drawing more meaningful conclusions. 257: related to language barriers [add “s”] 266: “host country’s language level”—I’m not sure what you mean, here. 275ff: Be careful of some claims here. For example, “Our study found that the type of service region influenced the overall socio-cultural adaptation and adaptation of every dimension”: But on Table 3, Class IV had a higher mean than non-hardship in social support and in interaction. 284: “Annual aid income influenced overall sociocultural adaptation, social morality, and social service.” Beyond the limitation that you note below, that your stats allow the measurement of correlation but not causality, based on F scores, you could say that annual aid influences (or rather, predicts) *every* aspect of sociocultural adaptation (all Fs are significant), though which means are highest differ for each dimension of sociocultural adjustment. Indeed, as I read Table 3, the highest level of national aid related (possibly) to the sub-area of social service. 292ff: Length of service: “Service periods of over 18 months demonstrated the best overall adaptation of the study”; but on Table 3, service periods of 6-12 and 12-18 months both had higher means than over 18 months (though means are close and probably not significantly different).

Note that if there are changes to the interpretation of the findings based on my notes here, that may lead to changes in the focus and content of some of the explanations in the discussion; however, if the claims remain unchanged, the discussion section does exactly what I would expect a discussion section to do.

Response 4: 

The logical idea of statistical analysis in this study is that we first summarize both the demographic characteristics and the scores of all dimensions of socio-cultural adaptation of the participants, and the frequencies with proportions and mean values with standard deviations are presented in Tables 1-2.

Then, we used t/ F tests to compare the differences for the scores of all dimensions of socio-cultural adaptation between different population subgroups.

In the final step, stepwise multiple linear regression models were performed to assess the potential related factors that might affect CMATMs’ socio-cultural adaptation in all dimensions.

Our multivariable models included all demographic variables as indicated in the Methods section. Models selection were automated since we employed the stepwise techniques (that’s why only these variables with statistical differences were reported).

B represents the coefficients of each argument in the regression equation, and because each argument has different ranges of values, B does not reflect the magnitude of the individual arguments affecting the dependent variables.

PS: if necessary, we need to rely on the standard coefficient, but this does not have any practical significance here, which is why we set dummy variables -- the pairwise comparison issues that the reviewer 1 cares about in ANOVA have been taken into account in our multi-regression models.

The interpretations of the models were actually dependent on B, which may conflict with unadjusted results from the single-factor analysis (Table 3), and this is fully understandable. In each model, multiple categories within one variable correspond to a unique specific reference group.

* Please note that we did this (Table 3) because it's a general statistical specification, besides, we prefer to show the specific scores of each variable population in detail to our readers. This means that this part is not mandatory. Because there may be collinearity between these variables, we did statistical modeling. The final interpretation is based on the results of the multi-factor models. Hence, our discussion is based on Table 4.

For other related issues in this section, thank you very much for carefully reviewing this paper, we corrected these points and carefully checked the MS:

We have added more detailed results in the results section (the interpretation of the table result data has been fully expressed).

217:We have cancelled the expression “Similar results were found for social support”.

219-220: We have deleted “the”.

224-227: We have made it clearer that in terms of length of service, those with a service period of over 18 months had better overall adaptation than those with a service period of ≤ 6 months.

257: We have added “s”.

266: It means: the expatriates’ proficiency in the host country’s language. We have revised the expression in the MS.

Point 5: 306: McCall and Hollenbeck: Not in reference list (I did not compare all citations against the reference—I was just curious and wanted to look this one up and realized that there is no corresponding reference number)

Response 5: We have added the reference.

Point 6:308: subconscious (1 word)  

Response 6: We have revised.

Point 7: 325: “should be from both sides”—great point! But clarify the wording. If you mean that both sides need to adjust, this is a solid point to make.

Response 7: We have revised and made it clearer: both side-the side of the host country and the side of the sending state. Regarding the side of the host country, the characteristics of the host country, the host country’s policy towards foreigners, and the social and cultural tolerance of the host country impact the overall socio-cultural adaptation.

Reviewer 3 Report

I am generally satisfied with the comments and revisions of the authors. My thanks for their clarification.

One minor point I would still like to suggest relates to my previous comment 3. It currently remains unclear whether the sample from the survey is compositionally similar to the research population, raising external validity questions. I recognize that it is not possible to address that now, but I think it has merit to mention this point for future inquiries in the Discussion section.

Author Response

Review Report (Round 2)

Manuscript ID:

ijerph-563015

Title:

"Socio-cultural Adaptation and Its Related Factors for Chinese Medical Aid Team Members (CMATMs) in Africa"

Dear Editors and Reviewers,

We very much appreciate the thoughtful and critical feedback from the reviewers again. We are delighted at your decision that we revise and resubmit this manuscript. The reviewers have requested some further clarification and revisions to our manuscript, which we have completed, and all changes have been highlighted for ready identification. In addition, we have addressed each of the comments from you specifically, and the following points are the replies to editor and each reviewer point by point.

* Please note that the page and line number may be changed after MS is changed.

We are looking forward to hearing from you soon.

Sincerely,

Authors

Response to Reviewer 2 Comments

Point 1: I am generally satisfied with the comments and revisions of the authors. My thanks for their clarification.

One minor point I would still like to suggest relates to my previous comment 3. It currently remains unclear whether the sample from the survey is compositionally similar to the research population, raising external validity questions. I recognize that it is not possible to address that now, but I think it has merit to mention this point for future inquiries in the Discussion section.

Response 1: This is an excellent suggestion. Really, it is! In fact, the information of medical team members is unknown, which is related to national policies, and the Chinese government does not have this information as well. We talked to our client (the National Health Commission of China) about this issue before we started and after we finished this project.

As you suggested, we have mentioned this point for future inquiries in the Limitation.